# Influence of Prior Influenza Vaccination on Current Influenza Vaccine Effectiveness in Children Aged 1 to 5 Years

**DOI:** 10.3390/vaccines9121447

**Published:** 2021-12-07

**Authors:** Kazuhiro Matsumoto, Wakaba Fukushima, Saeko Morikawa, Masashi Fujioka, Tohru Matsushita, Megumi Kubota, Yoshina Yagi, Yoshio Takasaki, Shizuo Shindo, Yuji Yamashita, Takato Yokoyama, Yumi Kiyomatsu, Satoshi Hiroi, Keiko Nakata, Akiko Maeda, Kyoko Kondo, Kazuya Ito, Tetsuo Kase, Satoko Ohfuji, Yoshio Hirota

**Affiliations:** 1Department of Public Health, Osaka City University Graduate School of Medicine, Osaka 545-8585, Japan; wakaba@med.osaka-cu.ac.jp (W.F.); maedaakiko@nifty.com (A.M.); kaz-ito@healthcare-m.ac.jp (K.I.); kasetetsuo143@msic.med.osaka-cu.ac.jp (T.K.); satop@med.osaka-cu.ac.jp (S.O.); 2Research Center for Infectious Disease Sciences, Osaka City University Graduate School of Medicine, Osaka 545-8585, Japan; 3Department of Virology, Osaka Institute of Public Health, Osaka 537-0025, Japan; morikawa@iph.osaka.jp (S.M.); hiroi@iph.osaka.jp (S.H.); knakata@iph.osaka.jp (K.N.); 4Fujioka Pediatric Clinic, Tondabayashi 584-0074, Japan; fjok@silver.ocn.ne.jp; 5Matsushita Kids’ Clinic, Kadoma 571-0030, Japan; matsushita@kids-clinic.com; 6Kubota Children’s Clinic, Osaka 544-0033, Japan; meg@ki.rim.or.jp; 7Yagi Pediatric Clinic, Yao 581-0871, Japan; yagi-ped@hera.eonet.ne.jp; 8Takasaki Pediatric Clinic, Fukuoka 819-0052, Japan; yositakasaki@nifty.com; 9Shindo Children’s Clinic, Fukuoka 814-0121, Japan; szshindo@nifty.com; 10Yamashita Pediatric Clinic, Itoshima 819-1112, Japan; yuji-yamashita@hyu.bbiq.jp; 11Yokoyama Children’s Clinic, Kasuga 816-0801, Japan; yokoyama_rnk2101@yahoo.co.jp; 12Kiyomatsu Pediatric Clinic, Fukuoka 814-0021, Japan; yumi-ped@kb3.so-net.ne.jp; 13Management Bureau, Osaka City University Hospital, Osaka 545-8586, Japan; kyou@med.osaka-cu.ac.jp; 14Healthcare Management, College of Healthcare Management, Miyama 835-0018, Japan; 15Clinical Epidemiology Research Center, SOUSEIKAI, Fukuoka 813-0017, Japan; hiro8yoshi@lta-med.com

**Keywords:** influenza, repeated vaccination, test-negative design

## Abstract

Background: Although annual influenza vaccination is an important strategy used to prevent influenza-related morbidity and mortality, some studies have reported the negative influence of prior vaccination on vaccine effectiveness (VE) for current seasons. Currently, the influence of prior vaccination is not conclusive, especially in children. Methods: We evaluated the association between current-season VE and prior season vaccination using a test-negative design in children aged 1–5 years presenting at nine outpatient clinics in Japan during the 2016/17 and 2017/18 influenza seasons. Children with influenza-like illness were enrolled prospectively and tested for influenza using real-time RT-PCR. Their recent vaccination history was categorized into six groups according to current vaccination doses (0/1/2) and prior vaccination status (unvaccinated = 0 doses/vaccinated = 1 dose or 2 doses): (1) 0 doses in the current season and unvaccinated in prior seasons (reference group); (2) 0 doses in the current season and vaccinated in a prior season; (3) 1 dose in the current season and unvaccinated in a prior season; (4) 1 dose in the current season and vaccinated in a prior season; (5) 2 doses in the current season and unvaccinated in a prior season, and (6) 2 doses in the current season and vaccinated in a prior season. Results: A total of 799 cases and 1196 controls were analyzed. The median age of the subjects was 3 years, and the proportion of males was 54%. Overall, the vaccination rates (any vaccination in the current season) in the cases and controls were 36% and 53%, respectively. The VEs of the groups were: (2) 29% (95% confidence interval: −25% to 59%); (3) 53% (6% to 76%); (4) 70% (45% to 83%); (5) 56% (32% to 72%), and (6) 61% (42% to 73%). The one- and two-dose VEs of the current season were significant regardless of prior vaccination status. The results did not differ when stratified by influenza subtype/lineage. Conclusion: Prior vaccination did not attenuate the current-season VE in children aged 1 to 5 years, supporting the annual vaccination strategy.

## 1. Introduction

Influenza causes mild-to-severe symptoms such as fever, myalgia, cough, sore throat, and nasal congestion. It is usually self-limiting but sometimes causes serious complications, which can lead to death. Influenza vaccination is one of the best interventions for the prevention of influenza and its complications. Annual influenza vaccination is usually recommended because of waning immunity and antigenic drift of the circulating strain [1]. However, if influenza vaccines are administered every year, the total amount of vaccines throughout a person’s life might be large, suggesting a non-negligible influence from a lifetime perspective. Additionally, concerns have been raised that repeated influenza vaccination may reduce vaccine effectiveness (VE). Many studies have assessed the influence of repeated vaccination on the current-season VE. Randomized trials from the 1970s and 1980s yielded inconsistent conclusions [2,3]. In 1999, a meta-analysis did not find evidence of a negative influence of prior vaccination [4]. More recent studies showed an attenuation of the current-season VE among those who received frequent vaccinations in the past [5,6]. Therefore, the effect of repeated vaccination is controversial. The variability of the results might be attributed to the different immunological statuses of the subjects. Although most subjects in these studies were children and adults, few studies have specifically assessed the influence of prior vaccination in children. For a better understanding of the rationale for repeated vaccination, we examined the influence of prior vaccination on the current-season VE among children aged 1 to 5 years during two consecutive influenza seasons (2016/17–2017/18) in Japan using a test-negative design.

## 2. Materials and Methods

### 2.1. Study Design, Setting, and Participants

This study was a multicentre case-control study (test-negative design) in which five paediatric clinics in Osaka prefecture and four paediatric clinics in Fukuoka prefecture in Japan participated during two consecutive influenza seasons (2016/17–2017/18). For each influenza season, we enrolled subjects for a maximum of nine weeks during the influenza epidemic period, with the number of reported influenza patients per sentinel being greater than five based on the surveillance data in each prefecture. The surveillance data were used as an alert system based on the number of cases reported in a given week per sentinel clinic and hospital in the National Epidemiological Surveillance of Infectious Diseases [7]. Children with an influenza-like illness (ILI) who attended a clinic were eligible if they were less than 6 years old and visited within 7 days of illness onset. The definition of ILI was fever ≥38.0 °C plus cough, sore throat, runny nose, and/or dyspnoea. We excluded children who already received anti-influenza drugs for the current ILI, experienced anaphylaxis related to the influenza vaccine, were admitted to a hospital due to the current ILI, were less than 6 months old as of September 1 before each season, were institutionalized, or who had lived in another prefecture. 

In each clinic, physicians chose any 3 days from each week and enrolled a maximum of 5 children per day. If the children were eligible, we recruited them consecutively to eliminate selection bias. Informed consent was obtained from the guardians of the children by the paediatricians of each clinic. This study was performed in accordance with the Declaration of Helsinki, and the study protocol was approved by the ethics committees at Osaka City University Graduate School of Medicine and Osaka Institute of Public Health (No. 4416, 3911, 2997). We guaranteed that the refusal to participate in a study or the decision to withdraw from the study would not bring any disadvantages. Every precaution was taken to protect the privacy of subjects, including anonymization of the data before analysis.

### 2.2. Information Collection

We collected information about the age, sex, influenza vaccination history since birth, medical comorbidities, attendance at a nursery school, siblings, the date of illness onset, number of outpatient visits in the last year, and influenza diagnosis in the prior season, using a self-administered questionnaire. When we could not obtain the necessary information, we used medical records. Medical comorbidities included respiratory, cardiovascular, renal, neurological, haematological, allergic, and congenital diseases as well as immunosuppressed states [5]. According to the research protocol approved by the ethics committee, all data were stored at the Department of Public Health, Osaka City University Graduate School of Medicine, and the principal investigator (W.F.) was responsible for managing the data.

### 2.3. Laboratory Testing

We obtained nasal aspirates using an 8fr JMS catheter^®^ and performed a real-time reverse transcription polymerase chain reaction (RT-PCR) assay. The specimens were tested by real-time RT-PCR at the Osaka Institute of Public Health according to the protocol used by the National Institute of Infectious Diseases [8,9].

### 2.4. Statistical Analyses

Subjects less than 1 year old were excluded because they could not be vaccinated in the prior season. Subjects were classified as cases or controls depending on the results of the real-time RT-PCR assay. The cases were subjects with positive results, and the controls were subjects with negative results. The characteristics and clinical symptoms of the cases and controls were examined by χ^2^ test or Wilcoxon rank sum test. Logistic regression was used to estimate the VE against laboratory-confirmed influenza by calculating the vaccination odds ratio in test-positive cases vs. test-negative controls and by using the following formula: VE = (1 − OR) × 100 (%). To match cases and controls within the same age (1-year interval), week, season, and clinic in the analysis, conditional logistic regression models with stratification by these factors were employed. In addition, we controlled for sex, siblings, attendance of nursery school, interval between illness onset and enrolment (0–2/≥3 days), medical comorbidities, number of outpatient visits (0–4/5–9/≥10 times) in the last year, and influenza diagnosis in the prior season. We estimated the VE against any influenza for the 2016/17 and 2017/18 seasons combined. 

In Japan, two doses of the vaccine are recommended for children aged 6 months to 12 years. Two 0.25 mL doses of the vaccine 2–4 weeks apart are recommended for children aged 6 months to 2 years, and two 0.5 mL doses 2–4 weeks apart are recommended for children aged 3–12 years. We categorized subjects by their recent vaccination history into six groups according to the current vaccination dose (0/1/2) and prior vaccination status (unvaccinated = 0 dose/vaccinated = 1 dose or 2 doses): (1) 0 doses in the current season and unvaccinated in a prior season (reference group); (2) 0 doses in the current season and vaccinated in a prior season; (3) 1 dose in the current season and unvaccinated in a prior season; (4) 1 dose in the current season and vaccinated in a prior season; (5) 2 doses in the current season and unvaccinated in a prior season, and (6) 2 doses in the current season and vaccinated in a prior season. We also included their vaccination statuses for the season before the prior season (no/yes) in the model as an adjusted variable and calculated the VE of each group. Since children aged 1 year old could not be vaccinated in the season before the prior season, we treated them as never vaccinated until the season before the prior season in the analysis. We conducted an additional analysis in which the subjects were stratified by prior vaccination status and estimated the VE for the current season. Statistical analyses were conducted using SAS version 9.4. A *p*-value <0.05 or a positive lower bound of the confidence interval (CI) for VE indicated statistical significance.

## 3. Result 

### 3.1. Characteristics of the Cases and Controls in This Study

We recruited 1007 participants in the 2016/17 season and 1015 participants in the 2017/18 season. After excluding 27 participants who were less than 1 year old at enrolment, a total of 799 cases and 1196 controls were analyzed. Among the cases, 71 (9%) had influenza A(H1N1)pdm, 408(51%) had influenza A(H3N2), 293 (37%) had influenza B(Yamagata), 25 (3%) had influenza B(Victoria), 1 was co-infected with A(H3N2) and A(H1N1)pdm, and 1 was co-infected with A(H3N2) and B(Yamagata). The median age of the subjects was 3 years old, and the proportion of males was 54%. The characteristics of the cases and controls are shown in Table 1. The cases were likely to be older, to have siblings, to attend nursery school, and to be diagnosed with influenza in the prior season compared with controls. The number of outpatient visits in the last year was lower among the cases than controls.

### 3.2. Clinical Symptoms of the Cases and Controls in This Study

Table 2 compares the clinical symptoms of the study subjects. The cases were more likely to suffer from fever (>39 °C), cough, and sore throat. However, the controls were more likely to suffer from runny nose and dyspnoea. The interval between illness onset and enrolment of the cases was shorter than that for the controls. 

### 3.3. Current Season VE against Any Influenza

The fully adjusted VE of the current season against any influenza was 57% (95% CI: 30% to 74%) for one dose and 51% (32% to 65%) for two doses (Table 3).

### 3.4. VE against Any Influenza According to the Current Vaccination Dose and Prior Vaccination Status

Table 4 shows the VE for each combination of current vaccination dose and prior vaccination status. Compared with the reference category, the one-dose, fully adjusted VE of the current season was statistically significant, regardless of prior vaccination status (53% (6% to 76%) in previously unvaccinated subjects and 70% (45% to 83%) in previously vaccinated subjects). We found the same results for the two-dose, fully adjusted VE of the current season (56% (32% to 72%) in previously unvaccinated subjects and 61% (42% to 73%) in previously vaccinated subjects). A forest plot of the strain-specific VE is shown in Figure 1. Prior vaccination did not reduce current-season VE against each strain. When stratifying by prior vaccination status, as shown in Table 5, the VE of the two-dose regimen was significant among previously unvaccinated and vaccinated subjects (60% (34% to 75%) and 75% (36% to 90%), respectively). 

## 4. Discussion

In this study, we estimated the current-season VE during two consecutive influenza seasons: the one-dose VE and two-dose VE were statistically significant. We also found that prior vaccination did not have a negative influence on the current-season VE. Similar findings were obtained when evaluating the VE for each strain.

Past epidemiological studies on the influence of prior vaccination have yielded inconsistent results. Some studies reported the negative effects of prior vaccination [5,6,10,11,12,13,14,15,16], and other studies found no evidence that prior vaccination attenuated the current-season VE [17,18,19]. Our study was consistent with these latter reports. In addition, one study showed that infrequent prior vaccinations (one or two prior doses in four prior seasons) kept or improved the protection of the current season vaccination but that frequent prior vaccinations (more than two prior doses) interfered with the current-season VE [20]. A test-negative case-control study of children aged 6 months to 15 years concluded that repeated vaccinations were associated with a reduced VE in the 2016/17 season [12]. Conversely, another test-negative case-control study of children aged 2 to 17 years did not indicate that prior-season vaccination attenuated the VE [21], indicating that the influence of prior vaccination was still inconclusive. Most studies included adolescents and adults, and few studies have evaluated the influence of prior vaccination among children alone. The heterogeneity of prior influenza vaccination effects is reasonable because the immunological background of the subjects depends on many factors, including age, region, season, antigenicity between vaccine strains and circulating strains, vaccination history, and past natural influenza infection. 

Several clinical and experimental studies have proposed possible immunologic explanations for the effect of prior influenza vaccination. Some clinical studies showed that repeated vaccinations reduced antibody responses and influenza-specific memory B cell responses to subsequent vaccination [6,22]. One study on adults aged 22 to 49 years suggested that pre-existing antibodies could bind to the hemagglutinin protein of the influenza vaccine to form antigen–antibody complexes, which might decrease the amount of antigen available to stimulate naïve or memory B-cells [23]. Another study showed that pre-existing immunity was negatively correlated with the induction of influenza-specific memory B-cells in vaccination among children who were previously vaccinated [24]. In addition, an animal study showed that repeated vaccinations reduced virus-specific CD8^+^ T cell responses [25]. A clinical study also demonstrated that annual vaccinations prevented the induction of virus-specific CD8^+^ T cell immunity in children [26]. These studies indicated that pre-existing immunity could hamper current-season vaccination. However, a recent epidemiological study found that prior influenza vaccinations improved poor immunogenicity among children aged 6 months to 3 years [27]. Younger children may need more exposure to prior influenza vaccinations to induce substantial immune responses to subsequent influenza vaccinations because their immune systems are more immature. In contrast, if immunity has already been established, increased vaccination stimulus might be harmful. Taken together, the cumulative lifetime exposure to influenza vaccination or natural infection may change the direction of the influence from prior vaccinations. In our study, prior vaccinations did not alter the current-season VE, probably because all subjects were children aged 1 to 5 years and their immune systems may be immature. Further studies providing a detailed explanation of the effect of prior vaccination when considering the immunological status of subjects are needed.

It has been suggested that subjects vaccinated during a prior season have a different vaccination and infection history from our reference group (i.e., 0 doses in the current season and unvaccinated in prior seasons) and, therefore, are likely to have different susceptibilities to influenza infections before vaccination, leading to a substantial bias of the VE estimate [28]. Therefore, we additionally estimated the VE stratified by prior vaccination status, a similar approach to that previously reported [13,29,30]. The VE of the two-dose regimen was significant, regardless of prior vaccination status, indicating that the current vaccine added protection to previously vaccinated and unvaccinated subjects.

This study had several limitations. First, we could not evaluate the influence of prior vaccination in B(Victoria) because of the small sample size. Additionally, the VE against A(H1N1)pdm had a wide CI. Second, we depended on information provided mainly from a self-administered questionnaire, which was not validated. Information that relies solely on a questionnaire might be compromised in terms of reliability and may cause misclassification. However, since the guardians of the subjects filled in the self-administered questionnaire before the examination for influenza, this misclassification was non-differential (bias about the effect of vaccination tended towards a null value). Third, the dosage of influenza vaccination was different between children aged less than 3 years and those aged 3 to 5 years (0.25 mL/dose and 0.5 mL/dose, respectively). Therefore, the dosage of prior vaccinations may also have been different between children aged 1 to 3 years and those aged 4 to 5 years, which may have affected our interpretation of the influence of prior vaccinations. Fourth, the cumulative experience of natural influenza infections might affect immunological status; however, we did not obtain precise information regarding this. Finally, our influenza vaccination practices are different from other countries because a two-dose vaccination is recommended for children every year in Japan. Therefore, caution should be applied when comparing our results with those from other countries.

## 5. Conclusions

We found that prior influenza vaccination was not associated with a reduced VE among children aged 1 to 5 years. This result supports the current recommendations for annual influenza vaccination of children.

## Figures and Tables

**Figure 1 vaccines-09-01447-f001:**
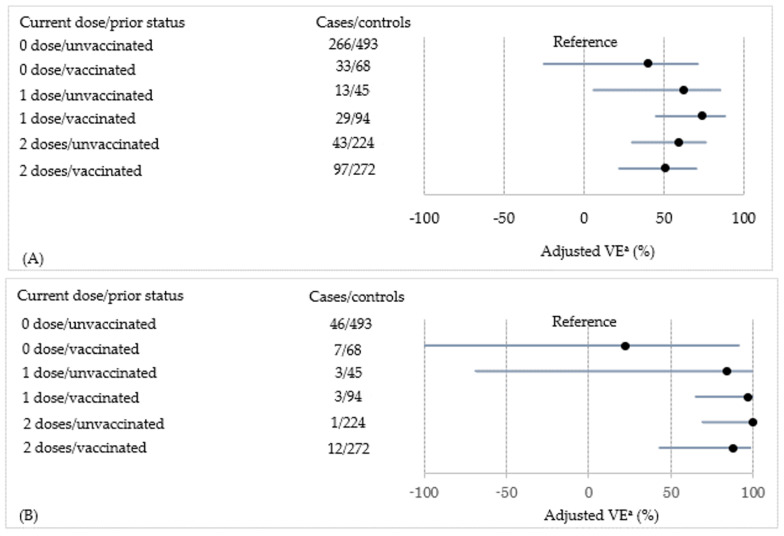
The adjusted vaccine effectiveness (VE) against each strain according to the current vaccination dose and prior vaccination status is shown as a forest plot. (**A**) Influenza type A, (**B**) A(H1N1)pdm, (**C**) A(H3N2), (**D**) type B, and (**E**) B(Yam). ^a^ Stratified variables: enrolment season, clinic, week, and age (1-year interval). Adjusted variables: sex, sibling, nursery school, interval between illness onset and enrolment (0–2/≥3days), medical comorbidities, number of outpatient visits (0–4/5–9/≥10) in the prior season, influenza diagnosis in the prior season, and influenza vaccination until the season before the prior season (0/≥1). ●: Point estimate of the adjusted VE. Black line indicates the 95% confidence interval of the adjusted VE. Note that we could not estimate the VE against B(Vic) because of the small sample size.

**Table 1 vaccines-09-01447-t001:** Characteristics of the study participants.

Variables	Cases (*n* = 799)	Controls (*n* = 1196)	*p*-Value ^a^
Male	430 (54)	652 (54)	0.76
Age (years)			
Median (range)	3 (1–5)	2 (1–5)	<0.01
Age (years)			
1	136 (17)	371 (31)	
2	143 (18)	290 (24)	
3	159 (20)	213 (18)	
4	191 (24)	196 (16)	
5	170 (21)	126 (11)	<0.01
Enrolment season			
2016/17	367 (46)	627 (52)	
2017/18	432 (54)	569 (48)	<0.01
Siblings (present)	641 (80)	835 (70)	<0.01
Nursery school (yes)	662 (83)	896 (75)	<0.01
Medical comorbidities ^b^	70 (9)	130 (11)	0.12
Number of outpatient visits in last year			
0–4	471 (59)	549 (46)	
5–9	209 (26)	367 (31)	
≥10	119 (15)	280 (23)	<0.01
Influenza vaccination in prior season	255 (32)	434 (36)	0.04
Vaccinated until the season before the prior season (+) ^c^	281 (35)	313 (26)	<0.01
Influenza diagnosis in prior season (+)	157 (20)	175 (15)	<0.01

Data are expressed as *n* (%) unless otherwise indicated. ^a^ Chi-square test or Wilcoxon rank sum test was used as appropriate. ^b^ Respiratory/heart/renal/neurology/hematology/allergic/congenital diseases and immunosuppression state. ^c^ All children aged 1 year old were treated as never vaccinated until the season before the prior season in the analysis.

**Table 2 vaccines-09-01447-t002:** Clinical symptoms of the study participants.

Variables	Cases (*n* = 799)	Controls (*n* = 1196)	*p*-Value ^a^
Maximum body temperature (°C)			
Median (range)	39.0 (38.0–41.0)	39.0 (38.0–41.5)	0.03
Maximum body temperature (°C)			
38.0–38.9	320 (40)	525 (44)	
≥39.0	479 (60)	671 (56)	0.09
Cough (+)	644 (81)	844 (71)	<0.01
Sore throat (+)	160 (20)	183 (15)	<0.01
Runny nose (+)	712 (89)	1124 (94)	<0.01
Dyspnea (+)	122 (15)	225 (19)	0.04
Interval between illness onset to enrolment (days)			
Median [range]	1 (0–6)	1 (0–7)	<0.01
0–2	762 (95)	1077 (90)	
≥3	37 (5)	119 (10)	<0.01

Data are expressed as *n* (%) unless otherwise indicated. ^a^ Chi-square test or Wilcoxon rank sum test were used as appropriate.

**Table 3 vaccines-09-01447-t003:** Current VE against any influenza.

Current Dose	Cases (*n* = 799)	Controls (*n* = 1196)	Crude VE (95% CI)	Adjusted VE (95% CI) ^a^	Adjusted VE (95% CI) ^b^
0	513 (64)	561 (47)	Reference	Reference	Reference
1	77 (10)	139 (12)	39% (18% to 55%)	66% (46% to 78%)	57% (30% to 74%)
2	209 (24)	496 (41)	54% (44% to 62%)	60% (47% to 70%)	51% (32% to 65%)

VE; vaccine effectiveness. CI; confidence interval. ^a^ Stratified variables: enrolment season, clinic, week, and age (1-year interval). ^b^ Stratified variables: enrolment season, clinic, week, and age (1-year interval). Adjusted variables: sex, sibling, nursery school, interval between illness onset and enrolment (0–2/≥3 days), medical comorbidities, number of outpatient visits (0–4/5–9/≥10) in the prior season, influenza vaccination in the prior season, and influenza diagnosis in the prior season.

**Table 4 vaccines-09-01447-t004:** VE against any influenza according to current and prior vaccination statuses.

Current Dose	Prior Vaccination	*n* (%)		Crude VE (95% CI)	Adjusted VE (95% CI) ^a^	Adjusted VE (95% CI) ^b^
		Cases (*n* = 799)	Controls (*n* = 1196)			
0	Unvaccinated	458 (57)	493 (41)	Reference	Reference	Reference
0	Vaccinated	55 (7)	68 (6)	13% (−27% to 40%)	27% (−25% to 57%)	29% (−25% to 59%)
1	Unvaccinated	25 (3)	45 (4)	40% (1% to 64%)	54% (12% to 77%)	53% (6% to 76%)
1	Vaccinated	52 (7)	94 (8)	41% (15% to 59%)	73% (53% to 85%)	70% (45% to 83%)
2	Unvaccinated	61 (8)	224 (19)	71% (60% to 79%)	61% (40% to 74%)	56% (32% to 72%)
2	Vaccinated	148 (19)	272 (23)	41% (26% to 54%)	62% (46% to 74%)	61% (42% to 73%)

VE; vaccine effectiveness. CI; confidence interval. ^a^ Stratified variables: enrolment season, clinic, week, and age (1-year interval). ^b^ Stratified variables: enrolment season, clinic, week, and age (1-year interval). Adjusted variables: sex, sibling, nursery school, interval between illness onset and enrolment (0–2/≥3 days), medical comorbidities, number of outpatient visits (0–4/5–9/≥10) in the prior season, influenza diagnosis in the prior season, and influenza vaccination until the season before the prior season (0/≥1).

**Table 5 vaccines-09-01447-t005:** VE against any influenza stratified by prior vaccination.

Current Vaccination Dose	*n* (%)		Crude VE (95% CI)	Adjusted VE (95% CI) ^a^	Adjusted VE (95% CI) ^b^
Unvaccinated in prior season	Cases (*n* = 544)	Controls (*n* = 762)			
0	458 (84)	493 (65)	Reference	Reference	Reference
1	25 (5)	45 (6)	40% (1% to 64%)	44% (−22% to 74%)	40% (−35% to 73%)
2	61 (11)	224 (29)	71% (60% to 79%)	64% (42% to 78%)	60% (34% to 75%)
Vaccinated in prior season	Cases (*n* = 255)	Controls (*n* = 434)			
0	55 (22)	68 (16)	Reference	Reference	Reference
1	52 (20)	94 (22)	32% (−12% to 58%)	76% (31% to 91%)	80% (36% to 94%)
2	148 (58)	272 (62)	33% (−1% to 55%)	69% (29% to 87%)	75% (36% to 90%)

VE; vaccine effectiveness. CI; confidence interval. ^a^ Stratified variables: enrolment season, clinic, week, and age (1-year interval). ^b^ Stratified variables: enrolment season, clinic, week, and age (1-year interval). Adjusted variables: sex, sibling, nursery school, interval between illness onset and enrolment (0–2/≥3 days), medical comorbidities, number of outpatient visits (0–4/5–9/≥10) in the prior season, influenza diagnosis in the prior season, and influenza vaccination until the season before prior season (0/≥1).

## Data Availability

The data presented in this study are available on request from the corresponding author. The data are not publicly available due to ethical reasons.

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
