# Peer review of "Influence of Prior Influenza Vaccination on Current Influenza Vaccine Effectiveness in Children Aged 1 to 5 Years"

_vaccines, 2021, doi:10.3390/vaccines9121447_

Round 1
Reviewer 1 Report
In this manuscript, Kazuhiro Matsumoto et al. assessed the impact of prior vaccination on current quadrivalent influenza vaccine effectiveness in children aged 1-5.
In the study, the authors recruited 799 children were infected with influenza virus and 1196 children with influenza like illness but were not infected with influenza virus based on RT-PCR diagnostics in seasons 2016/17 and 2017/18. All the children were categorized into 6 groups based on the vaccine status in current and prior seasons. Vaccine effectiveness was calculated based on the influenza specific RT-PCR diagnostics and compared among the 6 groups.
In all the analyses, the group of no vaccine in current or prior season was used as the reference for the other 5 groups. I wonder if the conclusion would be different if the group of no vaccine in current or prior season was used as the reference only for the 2 groups with no prior season vaccine; whilst the group of no vaccine in current season but vaccinated in prior season was used as the reference for the 2 groups with prior season vaccine.
The authors mentioned that children younger than 1 year old were excluded in the study since the prior season vaccine status was not applicable. In table 1 “vaccinated until the season before prior season” analysis, should all the children younger than 2 years old be excluded, since the season before prior season vaccine status was not applicable here?
In lines 284-286, the values from supposedly Table S1 are different from the values shown in Table 4. Table S1 is not included in the manuscript for revision. What are the differences in the analyses performed in Table 4 and Table S1?
In the abstract, the age information of the children included in the study was written as mean/average. However, in Table 1 the age was shown as median (range). Authors are invited to make this consistent across the entire manuscript.
Author Response
In all the analyses, the group of no vaccine in current or prior season was used as the reference for the other 5 groups. I wonder if the conclusion would be different if the group of no vaccine in current or prior season was used as the reference only for the 2 groups with no prior season vaccine; whilst the group of no vaccine in current season but vaccinated in prior season was used as the reference for the 2 groups with prior season vaccine.
In lines 284-286, the values from supposedly Table S1 are different from the values shown in Table 4. Table S1 is not included in the manuscript for revision. What are the differences in the analyses performed in Table 4 and Table S1?
Authors’ response
We really appreciate your important comment. Our Table S1 showed the results from exactly the perspective you pointed out. In the revised manuscript, Table S1 is presented as one of the main tables as Table 5 (Line 245-246). As shown in Table 5, the VE of the two dose regimen was significant regardless of prior vaccination. As we already stated, prior vaccination did not show a negative influence on current season VE in both analyses (Table 4,5). Thus, our conclusion was not different.
The authors mentioned that children younger than 1 year old were excluded in the study since the prior season vaccine status was not applicable. In table 1 “vaccinated until the season before prior season” analysis, should all the children younger than 2 years old be excluded, since the season before prior season vaccine status was not applicable here?
Authors’ response
Thank you for your comment. Since main interests in this study are current vaccination dose and prior vaccination status, we needed to exclude children younger than 1 year old to employ main analysis. Since "vaccinated until the season before prior season" was one of the confounders, we treated all the children aged 1 year old as those who had been “never vaccinated until the season before prior season" in the analysis. Therefore, we have added the annotation to the Materials and Methods (Lines 148-150) and footnote of Table 1 (Lines 202-203).
In the abstract, the age information of the children included in the study was written as mean/average. However, in Table 1 the age was shown as median (range). Authors are invited to make this consistent across the entire manuscript
Authors’ response
Thank you for your comment. We have corrected the mean age of the subjects to the median age in the Abstract (Line 41). We also have added the following sentence “The median age of the subjects was 3 years old and the proportion of males was 54%.”in the Result (Lines 163-164).
Reviewer 2 Report
I've been invited to review the paper entitled "Influence of prior influenza vaccination on current influenza vaccine effectiveness in young children" from the study group lead by Dr. Matsumoto Kazuhiro, from the Osaka City University.
Researchers have reported on a multicentre case-control study in which five paediatric clinics in Osaka prefecture, and 4 paediatric clinics in Fukuoka prefecture, participated during two consecutive seasons (2016/17-2017/18) to the vaccination campaign against Seasonal Influenza Vaccine (SIV) in order to understand whether vaccination against the previous seasonal influenza strain may influence or not eventual vaccination efficacy.
In the end, not only having received 2 doses in two consecutive years was associated with an improved VE, but repeated vaccination was unable to impair VE.
These results are quite interesting, and having been collected before the extensive, worldwide implementation of NPI against SARS-CoV-2, may be of substantial interest for the readers of Vaccines all around the world.
I've no substantial requirements in terms of statistical analysis or modelling of the paper: from this point of view, the paper may be accepted as it is. On the contrary, I would recommend the authors to double check the paper in terms of English phrasing, as some uncertainties (but also some minor mistakes) impair the direct understanding of the text.
Author Response
These results are quite interesting, and having been collected before the extensive, worldwide implementation of NPI against SARS-CoV-2, may be of substantial interest for the readers of Vaccines all around the world.
I've no substantial requirements in terms of statistical analysis or modelling of the paper: from this point of view, the paper may be accepted as it is. On the contrary, I would recommend the authors to double check the paper in terms of English phrasing, as some uncertainties (but also some minor mistakes) impair the direct understanding of the text.
Authors’ response
Thank you for your comment. Following your recommendation, we used English editing service again to check the paper in terms of English phrasing.
Reviewer 3 Report
The introduction of the manuscript is very brief. The context of the research is not clearly explored and there is no proper explanation why is this research needed. A stronger rationale is needed.
in methods, it is written seasons. What do you mean by seasons?
There is no proper ethical information included. How the central ethical issues were maintained?
What do you mean by young children? Kindly specifically mention the age group you are mainly targeting here.
Was there any validated questionnaire? If not, how the questionnaire was validated?
How the data was stored and who was responsible for it? Did you anonymise the data before analysis?
Author Response
The introduction of the manuscript is very brief. The context of the research is not clearly explored and there is no proper explanation why is this research needed. A stronger rationale is needed.
Authors’ response
We really appreciate your critical suggestions. Basically, influenza vaccination is common for all ages including children. When considering the possibility that the total amount of influenza vaccination in whole life may be large, it is important to clarity the rational of repeated vaccination. We have added the following sentences in Introduction, “However, if influenza vaccines are administered every year, the total amount of vaccines throughout a person’s life might be large, suggesting a non-negligible influence from a lifetime perspective.” (Lines 57-59) and “ For a better understanding of the rationale for repeated vaccination,” (Lines 69-70).
There is no proper ethical information included. How the central ethical issues were maintained?
Authors’ response
Thank you for your comment. We have added the phrases “by the paediatricians of each clinic” (Line 94) and the following sentences " We guaranteed that the refusal to participate in a study or the decision to withdraw from the study would not bring any disadvantages. Every precaution was taken to protect the privacy of subjects, including anonymization of the data before analysis.” in the Materials and Methods (Lines 97-100).
in methods, it is written seasons. What do you mean by seasons
Authors’ response
Thank you for your comment. We have added "influenza" to seasons in order to clarify the meaning of the term in Abstract (Line 33), Introduction (Line 71), Materials and Methods (Lines 78 and 79) and Discussion (Line 286).
What do you mean by young children? Kindly specifically mention the age group you are mainly targeting here.
Authors’ response
Thank you for your comment. We have deleted "young" and added "aged 1 to 5 years" in the Title, Abstract (Line 48) and Conclusion (Line 362).
Was there any validated questionnaire? If not, how the questionnaire was validated?
Authors’ response
Thank you for your comment. We did not use validated questionnaires to collect self-reported information, although they were structured. With regard to some information such as influenza vaccination history, we used medical record to validate self-reported information. We have revised the sentences in the limitation part as follows, “Second, we depended on information provided mainly from a self-administered questionnaire, which was not validated. Information that relies solely on a questionnaire might be compromised in terms of reliability and may cause misclassification.” (Lines 343-346).
How the data was stored and who was responsible for it? Did you anonymise the data before analysis?
Authors’ response
Thank you for your suggestion. We have added the following sentences " According to the research protocol approved by the ethics committee, all data were stored at the Department of Public Health, Osaka City University Graduate School of Medicine, and the principle investigator (W.F.) was responsible for managing the data.” in the Materials and Methods (Lines 109-112). As we explained earlier to protect the privacy of the subjects, we anonymized the data before analysis (Lines 99-100).
Round 2
Reviewer 1 Report
In the revised manuscript, the authors have addressed all the questions and comments from report 1.
Reviewer 3 Report
Thanks for making all the necessary changes and incorporating them into the manuscript effectively.